# A STRAIGHTFORWARD LINE SEARCH APPROACH ON THE EXPECTED EMPIRICAL LOSS FOR STOCHASTIC DEEP LEARNING PROBLEMS

## ABSTRACT

A fundamental challenge in deep learning is that the optimal step sizes for update steps of stochastic gradient descent are unknown. In traditional optimization, line searches are used to determine good step sizes, however, in deep learning, it is too costly to search for good step sizes on the expected empirical loss due to noisy losses. This empirical work shows that it is possible to approximate the expected empirical loss on vertical cross sections for common deep learning tasks considerably cheaply. This is achieved by applying traditional one-dimensional function fitting to measured noisy losses of such cross sections. The step to a minimum of the resulting approximation is then used as step size for the optimization. This approach leads to a robust and straightforward optimization method which performs well across datasets and architectures without the need of hyperparameter tuning.

## 1 INTRODUCTION AND BACKGROUND

The automatic determination of an optimal learning rate schedule to train models with stochastic gradient descent or similar optimizers is still not solved satisfactorily for standard and especially new deep learning tasks. Frequently, optimization approaches utilize the information of the loss and gradient of a single batch to perform an update step. However, those approaches focus on the batch loss, whereas the optimal step size should actually be determined for the empirical loss, which is the expected loss over all batches. In classical optimization line searches are commonly used to determine good step sizes. In deep learning, however, the noisy loss functions makes it impractically costly to search for step sizes on the empirical loss. This work empirically revisits that the empirical loss has a simple shape in the direction of noisy gradients. Based on this information, it is shown that the empirical loss can be easily fitted with lower order polynomials in these directions. This is done by performing a straightforward, one-dimensional regression on batch losses sampled in such a direction. It then becomes simple to determine a suitable minimum and thus a good step size from the approximated function. This results in a line search on the empirical loss. Compared to the direct measurement of the empirical loss on several locations, our approach is cost-efficient since it solely requires a sample size of about 500 losses to approximate a cross section of the loss. From a practical point of view this is still too expensive to determine the step size for each step. Fortunately, it turns out to be sufficient to estimate a new step size only a few times during a training process, which, does not require any additional time due to more beneficial update steps. We show that this straightforward optimization approach called ELF (Empirical Loss Fitting optimizer), performs robustly across datasets and models without the need for hyperparameter tuning. This makes ELF a choice to be considered in order to achieve good results for new deep learning tasks out of the box.

In the following we will revisit the fundamentals of optimization in deep learning to make our approach easily understandable. Following Goodfellow et al. (2016), the aim of optimization in deep learning generally means to find a global minimum of the true loss (risk) function $\mathcal{L}_{true}$ which is the expected loss over all elements of the data generating distribution $p_{data}$:

$$\mathcal{L}_{true}(\theta) = \mathbb{E}_{(\mathbf{x},y)\sim p_{data}} L(f(\mathbf{x};\theta), y) \tag{1}$$

where $L$ is the loss function for each sample $(\mathbf{x}, y)$, $\theta$ are the parameters to optimize and $f$ the model function. However, $p_{data}$ is usually unknown and we need to use an empirical approximation $\hat{p}_{data}$,

which is usually indirectly given by a dataset $\mathbb{T}$. Due to the central limit theorem we can assume $\hat{p}_{data}$ to be Gaussian. In practice optimization is performed on the empirical loss $\mathcal{L}_{emp}$:

$$\mathcal{L}_{emp}(\theta) = \mathbb{E}_{(\mathbf{x},y)\sim\hat{p}_{data}} L(f(\mathbf{x};\theta), y) = \frac{1}{|\mathbb{T}|} \sum_{(\mathbf{x},y)\in\mathbb{T}} L(f(\mathbf{x};\theta), y) \tag{2}$$

An unsolved task is to find a global minimum of $\mathcal{L}_{true}$ by optimizing on $\mathcal{L}_{emp}$ if $|\mathbb{T}|$ is finite. Thus, we have to assume that a small value of $\mathcal{L}_{emp}$ will also be small for $\mathcal{L}_{true}$. Estimating $\mathcal{L}_{emp}$ is impractical and expensive, therefore we approximate it with mini batches:

$$\mathcal{L}_{batch}(\theta, \mathbb{B}) = \frac{1}{|\mathbb{B}|} \sum_{(\mathbf{x},y)\in\mathbb{B}\subset\mathbb{T}} L(f(\mathbf{x};\theta), y) \tag{3}$$

where $\mathbb{B}$ denotes a batch. We call the dataset split in batches $\mathbb{T}_{batch}$.
We now can reinterpret $\mathcal{L}_{emp}$ as the empirical mean value over a list of losses $\mathbb{L}$ which includes the output of $\mathcal{L}_{batch}(\theta, \mathbb{B})$ for each batch $\mathbb{B}$:

$$\mathcal{L}_{emp}(\theta) = \frac{1}{|\mathbb{L}|} \sum_{\mathcal{L}_{batch}(\theta,\mathbb{B})\in\mathbb{L}} \mathcal{L}_{batch}(\theta, \mathbb{B}) \tag{4}$$

A vertical cross section $l_{emp}(s)$ of $\mathcal{L}_{emp}(\theta)$ in the direction $d$ through the parameter vector $\theta_0$ is given by

$$l_{emp}(s; \theta_0, d) = \mathcal{L}_{emp}(\theta_0 + s \cdot \mathbf{d}) \tag{5}$$

For simplification, we refer to $l$ as line function or cross section. The step size to the minimum of $l_{emp}(s)$ is called $s_{min}$.

Many direct and indirect line search approaches for deep learning are often applied on $\mathcal{L}_{batch}(\theta, \mathbb{B})$ (Mutschler & Zell (2020), Berrada et al. (2019), Rolinek & Martius (2018), Baydin et al. (2017),Vaswani et al. (2019)). Mutschler & Zell (2020) approximate an exact line search, which implies estimating the global minimum of a line function, by using one-dimensional parabolic approximations. The other approaches, directly or indirectly, perform inexact line searches by estimating positions of the line function, which fulfill specific conditions, such as the Goldberg, Armijo and Wolfe conditions (Jorge Nocedal (2006)). However, Mutschler & Zell (2020) empirically suggests that line searches on $\mathcal{L}_{batch}$ are not optimal since minima of line functions of $\mathcal{L}_{batch}$ are not always good estimators for the minima of line functions of $\mathcal{L}_{emp}$. Thus, it seems more promising to perform a line search on $\mathcal{L}_{emp}$. This is cost intensive since we need to determine $L(f(\mathbf{x};\theta_0 + s \cdot \mathbf{d}), y)$ for all $(\mathbf{x}, y) \in \mathbb{T}$ for multiple s of a line function. Probabilistic Line Search (PLS) (Mahsereci & Hennig (2017)) addresses this problem by performing Gaussian process regressions, which result in multiple one dimensional cubic splines. In addition, a probabilistic belief over the first (= Armijo condition) and second Wolfe condition is introduced to find good update positions. The major drawback of this conceptually appealing but complex method is, that for each batch the squared gradients of each input sample have to be computed. This is not supported by default by common deep learning libraries and therefore has to be implemented manually for every layer in the model, which makes its application impractical. Gradient-only line search (GOLS1) (Kafka & Wilke (2019)) pointed out empirically that the noise of directional derivatives in negative gradient direction is considerably smaller than the noise of the losses. They argue that they can approximate a line search on $\mathcal{L}_{emp}$ by considering consecutive noisy directional derivatives. Adaptive methods, such as Kingma & Ba (2014) Luo et al. (2019) Reddi et al. (2018) Liu et al. (2019) Tieleman & Hinton (2012) Zeiler (2012) Robbins & Monro (1951) concentrate more on finding good directions than on optimal step sizes. Thus, they could benefit from line search approaches applied on their estimated directions. Second order methods, such as Berahas et al. (2019) Schraudolph et al. (2007) Martens & Grosse (2015) Ramamurthy & Duffy (2017) Botev et al. (2017) tend to find better directions but are generally too expensive for deep learning scenarios.

Our approach follows PLS and GOLS1 by performing a line search directly on $\mathcal{L}_{emp}$. We use a regression on multiple $\mathcal{L}_{batch}(\theta_0 + s \cdot \mathbf{d}, \mathbb{B})$ values sampled with different step sizes $s$ and different batches $\mathbb{B}$, to estimate a minimum of a line function of $\mathcal{L}_{emp}$ in direction $\mathbf{d}$. Consequently, this work is a further step towards efficient steepest descent line searches on $\mathcal{L}_{emp}$, which show linear convergence on any deterministic function that is twice continuously differentiable, has a relative minimum and only positive eigenvalues of the Hessian at the minimum (see Luenberger et al. (1984)). The details as well as the empirical foundation of our approach are explained in the following.

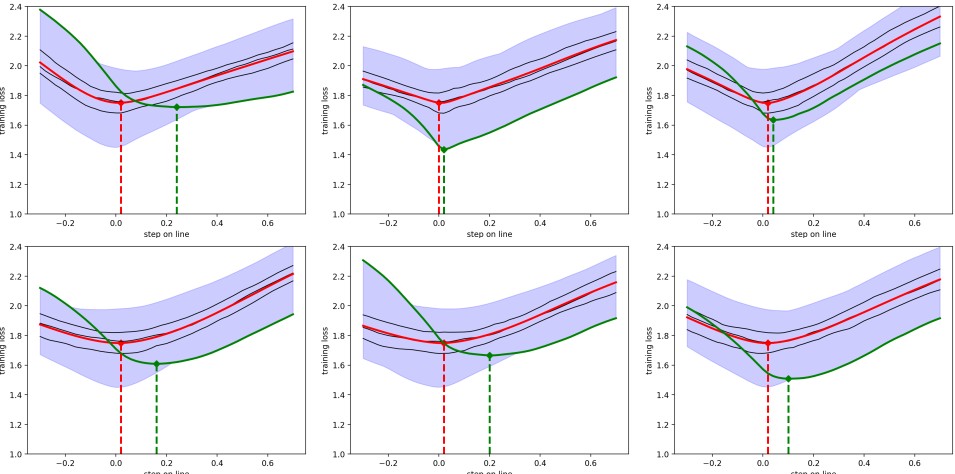

Figure 1: Distributions over all batch losses $\mathcal{L}_{batch}$ (blue) on consecutive and representative cross sections during a training process of a ResNet32 on CIFAR-10. The empirical loss $\mathcal{L}_{emp}$, is given in red, the quartiles in black. The batch loss, whose negative gradient defines the search direction, is given in green. See Section 2.1 for interpretations.

## 2 OUR APPROACH

### 2.1 EMPIRICAL FOUNDATIONS

Xing et al. (2018); Mutschler & Zell (2020); Mahsereci & Hennig (2017); Chae & Wilke (2019) showed empirically that line functions of $\mathcal{L}_{batch}$ in negative gradient directions tend to exhibit a simple shape for all analyzed deep learning problems. To get an intuition of how lines of the empirical loss in the direction of the negative gradient tend to behave, we tediously sampled $\mathcal{L}_{batch}(\theta_t + s \cdot -\nabla_{\theta_t}\mathcal{L}_{batch}(\theta_t, \mathbb{B}_t)), \mathbb{B})$ for 50 equally distributed $s$ between $-0.3$ and $0.7$ and every $\mathbb{B} \in \mathbb{T}$ for a training process of a ResNet32 trained on CIFAR-10 with a batch size of 100. The results are given in Figure 1. [1]

The results lead to the following characteristics: **1.** $l_{emp}$ has a simple shape and can be approximated well by lower order polynomials, splines or fourier series. **2.** $l_{emp}$ does not change much over consecutive lines. **3.** Minima of lines of $\mathcal{L}_{batch}$ can be shifted from the minima of $\bar{l}_{emp}$ lines and can even lead to update steps which increase $\mathcal{L}_{emp}$. Characteristic 3 consolidates why line searches on $\mathcal{L}_{emp}$ are to be favored over line searches on $\mathcal{L}_{batch}$. Although we derived these findings only from one training process, we can assure, by analyzing the measured point clouds of our approach, that they seem to be valid for all datasets, tasks, and models considered (see Appendix D).

### 2.2 OUR LINE SEARCH ON THE EXPECTED EMPIRICAL LOSS

There are two major challenges to be solved in order to perform line searches on $\mathcal{L}_{emp}$:

1. To measure $l_{emp}(s; \theta, \mathbf{d})$ it is required to determine every $L(f(\mathbf{x}; \theta_0 + s \cdot \mathbf{d}), y)$ for all $(\mathbf{x}, y) \in \mathbb{T}$ for all steps sized $s$ on a line.

2. For a good direction of the line function one has to know
$\nabla_\theta \mathcal{L}_{emp}(\theta) = \frac{1}{|\mathbb{T}|} \sum_{\mathbb{B} \in \mathbb{T}} \nabla \mathcal{L}_{batch}(\theta, \mathbb{B})$.

We solve the first challenge by fitting $l_{emp}$ with lower order polynomials, which can be achieved accurately by sampling a considerably low number of batch loss values. We do not have an efficient solution for the second challenge, thus we have to simplify the problem by taking the unit gradient of the current batch $\mathbb{B}_t$, which is $\hat{\nabla}_\theta \mathcal{L}_{batch}(\theta, \mathbb{B}_t)$, as the direction of the line search. The line function we search on is thus given by:

$$l_{ELF}(s; \theta_t, \mathbb{B}_t) = \mathcal{L}_{emp}(\theta_t + s \cdot -\hat{\nabla}_{\theta_t}\mathcal{L}_{batch}(\theta_t, \mathbb{B}_t)) \approx \text{lower order polynomial} \tag{6}$$

Note that $\theta_t, \mathbb{B}_t$ are fixed during the line search.

---

[1] These results have already been published by the authors of this paper in another context in [ref]

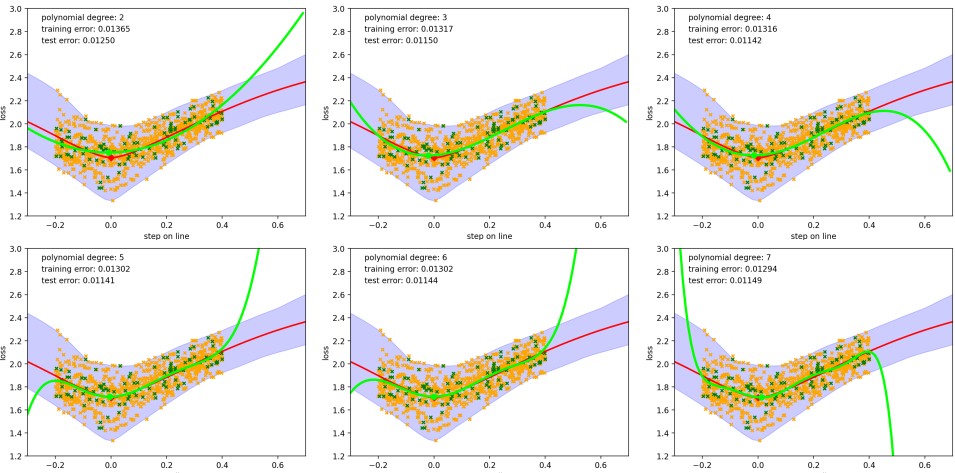

Figure 2: An exemplar ELF routine testing for the best fitting polynomial. The empirical loss in given in red, the distribution of batch losses in blue. The sampled losses are given in orange and green. The green losses are the test set of the current cross validation step. It can be seen, that the fifth-order polynomial (green) reaches the lowest test error.

Our straightforward concept is to sample $n$ losses $\mathcal{L}_{batch}(\theta_0 + s_i \cdot -\hat{\nabla}_\theta \mathcal{L}_{batch}(\theta, \mathbb{B}_0), \mathbb{B}_i)$, with $i$ ranging from 1 to $n$ and $\mathbb{B}_i$ uniformly chosen from $\mathbb{T}$ and $s_i$ uniformly chosen from a reasonable interval, on which we will focus later. Now, we follow a classical function fitting or machine learning routine. An ordinary least square regression (OLSR) for polynomials is performed. Note, that our data is not homoscedastic, as required for OLSR [2]. This implies, that our resulting estimator is still unbiased, but we cannot perform an analysis of variances (see Goldberger et al. (1964)). However, the latter is not needed in our case. Those regressions are performed with increasing degree until the test error of the fitted polynomial is increasing. The test error is determined by a 5-fold cross-validation. The second last polynomial degree is chosen and the polynomial is again fitted on all loss values to get a more accurate fit. Consequently the closest minimum to the initial location is determined and additional losses are measured in a reasonable interval around it. This process is repeated four times. Finally, the step to the closest minimum of the fitted polynomial is chosen as update step if, existing and its value is positive. Otherwise, a new line search is started. This routine is described in more detail in Algorithm 1. An empirical example of the search of the best fitting polynomial is given in Figure 2.2. The empirically found heuristic to determine reasonable measure intervals is given in Algorithm 2. This routine empirically ensures, that the point cloud of losses is wider than high, so that a correctly oriented polynomial is fitted. To determine when to measure a new step size with a line search, we utilize that one can estimate the expected improvement by $l_{ELF}(0) - l_{ELF}(s_{min})$. If the real improvement of the training loss times a factor is smaller than the expected improvement, both determined over a step window, a new step size is determined. The full ELF algorithm is given in Algorithm 3 in Appendix A. We note that all of the mentioned subroutines are easy to implement with the Numpy Python library, which reduces the implementation effort significantly. The presented pseudo codes include the most important aspects for understanding our approach. For a more detailed description we refer to our implementation found in the supplementary material.

Based on our empirical experience with this approach we introduce the following additions: **1.** We measure 3 consecutive lines and take the average resulting step size to continue training with SGD. **2.** We have experienced that ELF generalizes better if not performing a step to the minimum, but to perform a step that decreased the loss by a decrease factor $\delta$ of the overall improvement. In detail, we estimate $x_{target} > x_{min}$, which satisfies $f(x_{target}) = \delta(f(x_0) - x_{min}) - f(x_{min})$ with $\delta \in [0, 1)$ **3.** We use a momentum term $\beta$ on the gradient, which can lead to an improvement in generalization. **4.** To prevent over-fitting, the batch losses required for fitting polynomials are sampled from the validation set. **5.** At the beginning of the training a short grid search is done to find the maximal step size that still supports training. This reduces the chances of getting stuck in a local minima at the beginning of optimization.

---

[2]We indirectly use weighted OLSR by sampling more points in relevant intervals around the minimum, which softens the effect of heteroscedasticity.

---

**Algorithm 1** Pseudo code of ELF's line search routine (see Algorithm 3)

---
**Input:** d: direction (default: current unit gradient)
**Input:** $\theta_0$: initial parameter space position
**Input:** $\mathcal{L}_{batch}(\theta_t)$: batch loss function which randomly chooses a batch
**Input:** $k$: sample interval adaptations (default: 5)
**Input:** $n$: losses to sample per adaptation (default: 100)
 1: interval_width $\leftarrow$ 1.0
 2: sample_positions $\leftarrow$ []
 3: lineLosses $\leftarrow$ []
 4: **for** $r$ from 0 to k **do**
 5:    **if** r != 0 **then**
 6:       interval_width $\leftarrow$ chose_sample_interval(minimum_location, sample_positions, line_losses, coefficents)
 7:    **end if**
 8:    new_sample_positions $\leftarrow$ get_uniformly_distributed_values(n, interval_width)
 9:    **for** $m$ in new_sample_positions **do**
10:       line_losses.append($\mathcal{L}_{batch}(\theta_0 + m\mathbf{d})$)
11:    **end for**
12:    sample_positions.extend(new_sample_positions)
13:    last_test_error $\leftarrow \infty$
14:    **for** degree from 0 to max_polynomial_degree **do**
15:       test_error $\leftarrow$ 5-fold_cross_validation(degree, sample_positions, line_losses)
16:       **if** last_test_error $<$ test_error **then**
17:          best_degree $\leftarrow$ degree$-1$
18:          last_test_error $\leftarrow$ test_error
19:          **break**
20:       **end if**
21:       **if** degree == max_polynomial_degree **then**
22:          best_degree $\leftarrow$ max_polynomial_degree
23:          **break**
24:       **end if**
25:    **end for**
26:    coefficients $\leftarrow$ fit_polynomial(best_degree, sample_positions, line_losses)
27:    minimum_position, improvement $\leftarrow$ get_minimum_position_nearest_to_0(coefficients)
28: **end for**
29: **return** minimum_position, improvement, $k \cdot n$

---

**Algorithm 2** Pseudo code of the chose_sample_interval routine of Algorithm 1

---
**Input:** minimum_position
**Input:** sample_positions
**Input:** line_losses : list of losses corresponding to the sample_positions
**Input:** coefficents : coefficients of the polynomial of the current fit
**Input:** min_window_size (default: 50)
 1: window $\leftarrow \{m : m \in$ sample_positions and $0 \leq m \leq 2 \cdot$ minimum_location$\}$
 2: **if** $|window| <$ min_window_size **then**
 3:    window $\leftarrow$ get_50_nearest_sample_pos_to_minimum_pos-
                (sample_positions, minimum_position)
 4: **end if**
 5: target_loss $\leftarrow$ get_third_quartile(window, line_losses[window])
 6: interval_width $\leftarrow$ get_nearest_position_where_the_absolut_of_polynomial-
                _takes_value(coefficents,target_loss)
 7: **return** interval_width

---

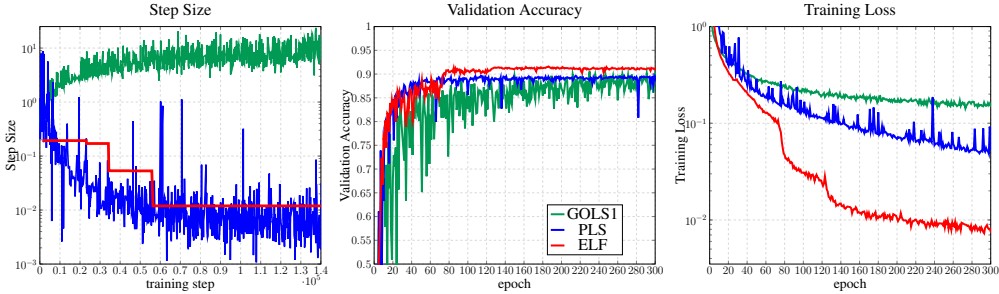

Figure 3: Comparison of ELF against PLS and GOLS1 on a ResNet-32 trained on CIFAR-10. The corresponding test accuracies are: EFL: 0.900, PLS: 0.875, GOLS1: 0.872. ELF performs better in this scenario and, intriguingly, PLS and ELF estimate similar step size schedules.

## 3 EMPIRICAL ANALASYS

To make our analysis comparable on steps and epochs, we define one step as loading of a new input batch. Thus, the steps/batches needed of ELF to estimate a new line step are included.

### 3.1 COMPARISON TO OPTIMIZATION APPROACHES OPERATING ON $\mathcal{L}_{emp}$

We compare against PLS (Mahsereci & Hennig, 2017) and GOLS1 (Kafka & Wilke, 2019). Both approximate line searches on the empirical loss. Since PLS has to be adapted manually for each Model that introduces new layers, we restrict our comparison to a ResNet-32 (He et al. (2016)) trained on CIFAR-10 (Krizhevsky & Hinton (2009)). In addition, we use an empirically optimized and the only available Tensorflow (Abadi et al. (2015)) implementation of PLS (Balles (2017)). For each optimizer we tested five appropriate hyperparameter combinations, which are likely to result in good results (see Appendix B). The best performing runs are given in Figure 3.1. We can see that ELF slightly surpasses GOLS1 and PLS on validation accuracy and training loss in this scenario. Intriguingly, PLS and ELF estimate similar step size schedules, whereas, that of GOLS1 differs significantly.

### 3.2 ROBUSTNESS COMPARISON OF ELF, ADAM AND SGD

Since ELF, as long as loss lines are well approximate-able by polynomials, should adapt to different loss landscapes, it is expected to perform robustly across models and datasets. Therefore, we will concentrate on robustness in this evaluation. For CIFAR-10, CIFAR-100, SVHN we will train on DenseNet-121, MobileNetV2, ResNet-18 and ResNet-34. For Fashion-MNIST we consider a 3-layer fully connected network and a 3-layer convolutional network. For ImageNet we consider ResNet-18 and MobileNetV2. We compare against the most widely used optimizer SGD, ADAM (widely considered to be robust (Schmidt et al., 2020; Kingma & Ba, 2014)) and PAL (Mutschler & Zell (2020)), a line search approach on $\mathcal{L}_{batch}$. At first, we perform an optimizer-hyperparameter grid search over the models considered for CIFAR-10. Then, we choose the hyperparameter combination that achieves on average the best test accuracy for each optimizer. Consequently robustness is evaluated by reusing those hyperparameters on each additional dataset and models. For ADAM and SGD we use a standard learning rate schedule, which divides the initial learning rate by 10 after half and after three quarters of the training steps. For ADAM and SGD we considered $10^{-1}, 10^{-2}, 10^{-3}, 10^{-4}$ as learning rates. ADAM's moving average factors are set to $0.9$ and $0.999$. For SGD, we used a momentum of $0.9$. For ELF we used the default value for each hyperparameter, except for the momentum factor, which was chosen as $0$ or $0.4$ and the decrease factor $\delta$ was chosen as $0$ or $0.2$. For PAL a measuring step size of $0.1$ or $1$ and a step size adaptation of $1.66$ and $1.0$ was considered. Further details are given in Appendix B.

As shown in Figure 4 our experiments revealed that the most robust hyperparameter combination is a momentum factor of $0.4$ and a $\delta$ of $0.2$ for ELF. For SGD, the most robust learning rate is $0.01$ and for ADAM $0.001$. For PAL a measuring step size of $0.1$ and a step size adaptation $1.0$ perform most robustly.

For all optimizers the most robust hyperparameters found on CIFAR-10 tend to perform also robustly on other datasets and models. This is unexpected, since it is generally assumed that a new learning

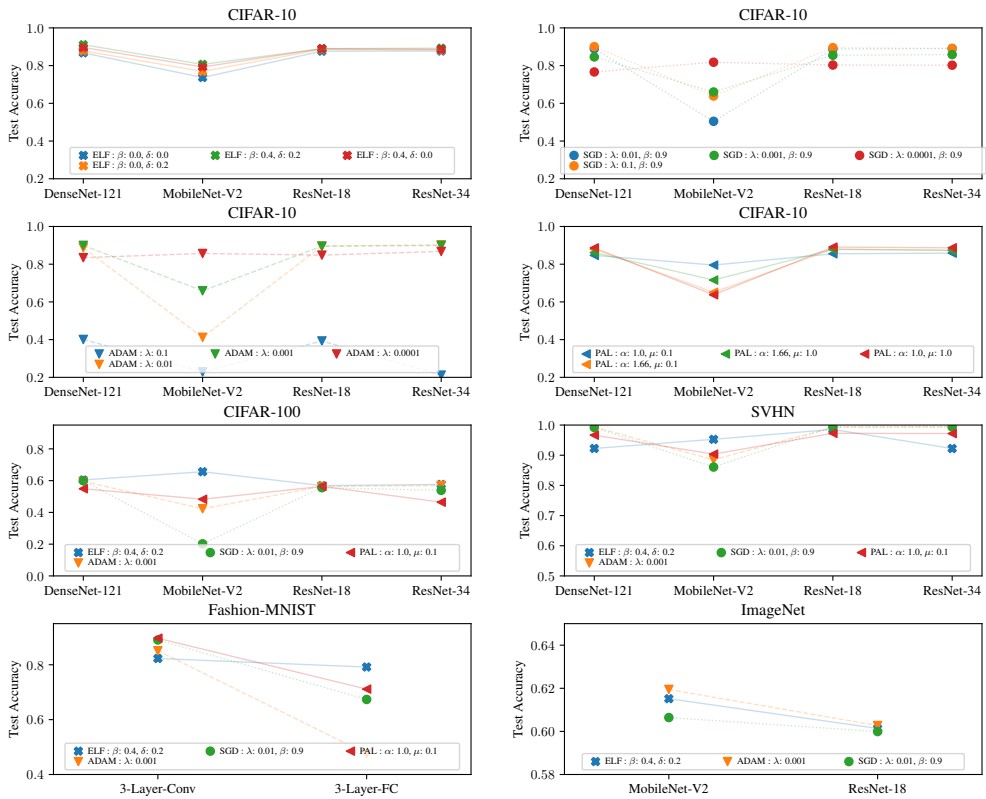

Figure 4: Robustness comparison of ELF, ADAM , SGD and PAL. The most robust optimizer-hyperparameters across several models were determined on CIFAR-10, which then tested on further datasets and models. On those, the found hyperparameters of ELF, ADAM, SGD and PAL behave robust. ($\beta$=momentum, $\lambda$=learning rate, $\delta$=decrease factor,$\alpha$=update step adaptation, $\mu$=measuring step size). Plots of the training loss and validation accuracy are given in Appendix C. (Results for PAL on ImageNet will be included in the final version.)

rate has to be searched for each new problem. In addition, we note that ELF tends to perform better on MobileNet-V2 and the 3-Layer-FC network, however, is not that robust on SVHN. The most important insight, which has been obtained from our experiments, is that for all tested models and dataset ELF was able to fit lines by polynomials. Thus, we are able to directly measure step sizes on the empirical loss. Furthermore, we see that those step sizes are useful to guide optimization on subsequent steps. To strengthen this statement, we plotted the sampled losses and the fitted polynomials for each approximated line. Representative examples are given in Figure 5 and in Appendix D.

Figure 6 (left) shows that ELF uses depending on the models between $1\%$ to $75\%$ of its training

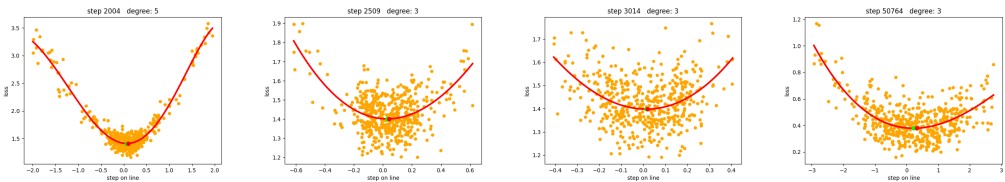

Figure 5: Representative polynomial line approximations (red) obtained by training ResNet-18 on CIFAR-10. The samples losses are depicted in orange. The minimum of the approximation is represented by the green dot, whereas the update step adjusted by a decrease factor of 0.2 is depicted as the red dot.

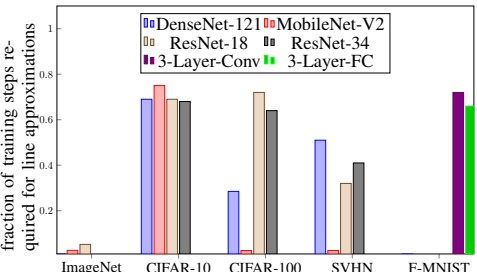 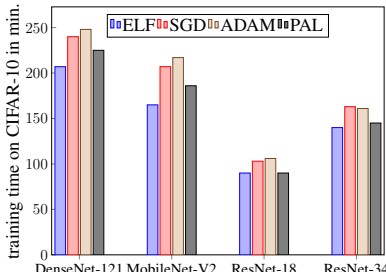

Figure 6: Left: Fraction of training steps used for step size estimation and the total amount of training steps. The amount of steps needed depends strongly on the model and dataset. Right: Training time comparison on CIFAR-10. One can observe, that ELF performs slightly faster.

steps to approximate lines. This shows, that depending on the problem more or less step sizes have to be determined. In addition, this indicates, that update steps become more efficient for models, for which many new step sizes had to be determined. Figure 6 (left) shows that ELF is faster than the other optimizers. This is a consequence of sole forward passes required for measuring the losses on lines and since the operations required to fit polynomials are cheap.

## 4 DISCUSSION & OUTLOOK

This work demonstrates that line searches on the true empirical loss can be done efficiently with the help of polynomial approximations. These approximations are valid for all investigated models and datasets. Although, from a practical point of view, measuring a new step size is still expensive, it seems to be sufficient for a successful training process to measure only a few exact step sizes during training. Our optimization approach ELF performs robustly over datasets and models without any hyperparameter tuning needed and competes against PAL, ADAM and SGD. The later 2 were run with a good learning rate schedule. In our experiments ELF showed better performance than Probabilistic Line Search and Gradient-only Line Search. Both also estimate their update steps from the expected empirical loss.

An open question is to what extent our approach leads to improvements in theory. It is known that exact steepest descent line searches on deterministic problems (Luenberger et al. (1984), p. 239)) exhibit linear convergence on any function, which is twice continuously differentiable and has a relative minimum at which the hessian is positive definite. The question to be considered is how the convergence behavior of an exact line search behaves in noisy steepest directions. This, to the best of our knowledge, has not yet been answered.

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
