# OpenReview forum: "A straightforward line search approach on the expected empirical loss for stochastic deep learning problems"
_ICLR.cc/2021/Conference — Reject_

### Official Review · AnonReviewer4 · 2020-10-23
**Proposes heuristic to tune step sizes in SGD during the training run, by sampling a number of extra batch losses now and then.**

**Rating:** 4
**Confidence:** 4

**Review:**

The paper tackles an important issue, namely how to tune the step sizes in SGD during the training, trying to approximate step sizes which would be used in GD (even though the search direction is still noisy). Relevant prior work is cited. The method is simple and easy to understand. Step sizes are kept piecewise constant over updates. New batches are sampled, and the loss values along the search line are fitted with a low-order polynomial. There is some heuristic to choose the interval around 0 for the step. Importantly, these batches are sampled from the validation set. While this sounds very elaborate to find the minimum of the approximation, they later say they are just trying for sufficient descent along the line. Compared to previous work, the method is simple and seems quite robust. As drawbacks, the method seems pretty expensive, and it has a large number of free parameters that need to be chosen.

A major weakness of the paper is the empirical evaluation. The comparison to related work in 3.1 is a bit meaningless, because they look at validation error. Now, their method uses the validation set extensively (to sample batches), while the others do not, so the comparison is flawed. One would have to evaluate on an independent test set. What is also really missing here are figures about the extra amount of time required. My suspicion is that PLS is quite a bit cheaper than what they do here, even though admittedly it needs specific implementations (which, for TensorFlow, are provided by the authors).
Then, in 3.2, they do not compare against this related work anymore. Why not? Again, I am missing a proper quantification of extra cost. It is also unclear how the many free parameters of their heuristic are chosen. For Figure 6 left, it is hard to understand why there is this difference. The reason is buried in the text somewhere: "To determine whether to measure a new step size...". This describes a rule for how often to update step size, and it again has unspecified free parameters. For Figure 6 right: This comparison is meaningless unless the stopping criteria for all methods are clearly stated. If ELF is using *extra* batch evaluations to adjust learning rate, it must be slower than ADAM or SGD if the same number of epochs are done. What takes the extra time for ADAM and SGD? Do they run more epochs? If so, what is the stopping rule? If that rule depends on validation error, the comparison is flawed (see above), because ELF accesses the validation set a lot, while ADAM and SGD do not (except for initial HPO). Or do they all run the same number of epochs? If so, what takes the extra time for ADAM and SGD? If this is some initial HPO, it has to be clearly specified, as it could range from cheap to very expensive depending on what is done.

Given these issues, I feel the work cannot be properly evaluated. My recommendation is to (a) clearly quantify the extra cost for all methods compared, (b) to compare against PLS and GOLS1 everywhere, (c) to be very specific about how training is stopped when comparing training runtime, and (d) to evaluate metrics on a dataset that ELF has no access to.

Since ELF seems quite expensive, it would be useful to try and see what can be done with batch losses evaluated along the training trajectory, instead of having to sample independent ones. They should also test how few batch losses are needed to still get good behaviour. In general, some ablation studies are needed that would drive down extra compute and check how fast this can be made.

---

> ### Author Response · Authors · 2020-11-17
> **Detailed reply to reviewer 4**
>
>
> **Dear reviewer 4 thanks again for the detailed and good feedback!**
> As described below we will follow your advice and will redo some experiments, and we will add additional ones.
>
> *"The comparison to related work in 3.1 is a bit meaningless, because they look at validation error. Now, their method uses the validation set extensively (to sample batches), while the others do not, so the comparison is flawed"*
> * Thus, we added the test errors in the caption of Figure 3, which still shows to be better. Unfortunately plotting the test error over time should not be done. We will redo this experiment without looking at the validation error.
>
> *"I am missing a proper quantification of extra cost. It is also unclear how the many free parameters of their heuristic are chosen"*
> * The empirical quantification of extra costs is given in Figure 5.
> * The intrinsic hyperparameters are searched by try and error until we were able to fit the empirical loss robustly for a ResNet18 and a DenseNet121 trained on Cifar10.
>
>
> *"Then, in 3.2, they do not compare against this related work anymore. Why not? "*
> * (Mutschler et al, 2020) showed empirically that (Vaswani et al., 2019) and (Berrada, et al) behave non robust on only slightly different problems. Anyways we will include a comparison against those.
>
>
> *"It is also unclear how the many free parameters of their heuristic are chosen"*
> * The used value for each hyperparameter can be found as default values in Algorithm 1,2,3
>
>
> *" This comparison is meaningless unless the stopping criteria for all methods are clearly stated"*
> * The stopping criteria is, unfortunately, stated in Appendix B. All algorithms were trained the same amount of steps.
> One step is considered as the loading and handling of one batch, as, described in the first sentence of section 3.
> * ELF performs faster than ADAM and SGD, since a large amount of "steps" is used to estimate new step sizes with line searches.
> 	For those loaded batches only a forward pass instead of a forward and a backward pass is performed. Similarly, for PAL only at every second step a backward pass is performed.
>
>
> *" They should also test how few batch losses are needed to still get good behaviour. In general, some ablation studies are needed that would drive down extra compute and check how fast this can be made"*
> * This is a crucial point, we will do such studies.
>
>
> *"It would be useful to try and see what can be done with batch losses evaluated along the training trajectory, instead of having to sample independent ones"*
> * We thought a lot about this but have not found any reasonable idea how to do this. It might help to estimate the expected loss at the current position and the norm of the exact gradient. However, we know no way to exploit this information for optimization.

---

> > ### Comment · AnonReviewer4 · 2020-11-18
> > **Response to author feedback**
> >
> > I read the author feedback, it does not change my vote. In fact, they do not really address my main points:
> > - Why not compare against the methods in 3.1 further down
> > - Comparison in 3.1 is broken
> > - How to set the very many free parameters on a new problem
> > - Why this method is even faster than SGD, given that SGD needs to do less. It can only be that their ELF converges faster, but that needs a proper convergence criterion

---

> > > ### Author Response · Authors · 2020-11-19
> > > **Response to reviewer4 feedback**
> > >
> > > We read the reviewers feedback and agree with his vote. However, we have to state that we indeed addressed most of his main points. We agree with him on some of them and will update our work as soon as possible.
> > > We note, that there is also a "General reply to all reviewers " above.
> > >
> > > *Why not compare against the methods in 3.1 further down*
> > > * We agreed that this should be done.
> > >
> > > *Comparison in 3.1 is broken*
> > > * We agreed and will redo this comparison.
> > >
> > > *How to set the very many free parameters on a new problem*
> > > * We have understood this one wrong. This is what the whole robustness evaluation of 3.2 is about. One just does not have to reset them. They are good enough for a wide range of problems.
> > >
> > > *Why this method is even faster than SGD, given that SGD needs to do less. It can only be that their ELF converges faster, but that needs a proper convergence criterion*
> > > * As explained in the comment above ELF does not need to perform a backward pass for each step, whereas SGD needs to and we used the same amount of steps for each optimizer.

---

### Official Review · AnonReviewer2 · 2020-10-27

**Rating:** 4
**Confidence:** 4

**Review:**

Update: I thank the authors for the detailled response. Due to the number of required changes and the feedback of other reviewers, I believe the paper needs a major revision before publication and still recommend rejection.


---

The submission introduces a heuristic to select the step-size for training deep learning models. Reducing the dependence on hyperparameters and improving the performance of optimization methods with out-of-the-box settings is an important problem and relevant to the ICLR community. The proposed approach is novel and well illustrated.

However, as there are limited theoretical contributions or new insights, the submission has to be evaluated on the effectiveness of the method. The main weakness of the manuscript is a lack of motivation for the details of the proposed heuristics and insufficient experimental evaluation.

Despite the method being described as simple and straightforward, I fail to see how the proposed approach is simpler than a backtracking line-search on the batch loss. A main argument for the proposed approach in constrast to previous work is that the minibatch loss is not sufficiently representative for a line-search. This contrasts with observations in previous work (e.g. Vaswani et al., 2019). While it makes intuitive sense that taking multiple batches into account is necessary in the presence of noise, there is limited evidence that the proposed method does so appropriately. The experimental evidence in section 3.1 is promising but is not broad enough to be informative. While the results of section 3.2 are nice to have, the improved robustness of the method against SGD and Adam is not surprising; the space would be better used comparing the performance of the method against the other cited works on batch line search.

My recommendation is towards a rejection. There is evidence the proposed method can work. But I doubt the submission will generate enthusiasm from the theory side of the community do to the limited new insights and the empirical evaluation is insufficient to convince me of the benefits of the method over previous work for its applications.


As additional feedback (there to help, not necessarily part of the decision), I advise the authors to pay a greater attention to presentation details when preparing their submissions. The writing and quality of the figures gets in the way of the message they are trying to communicate. For a few example, since the "homoscedasticity" is irrelevant for the current problem, I do not follow why it is mentioned so prominently in section 2. The use of "vertical cross-section" is also confusing; it is still unclear to me what is
vertical.

---

> ### Author Response · Authors · 2020-11-17
> **Detailed reply to reviewer 2**
>
>
> **Dear reviewer 2 thanks again for the detailed and good feedback!**
> As described below we will follow your advice and will redo some experiments, and we will add additional ones.
> Note, that there is also a "General reply to all reviewers " above.
>
>
> "*The main weakness of the manuscript is a lack of motivation for the details of the proposed heuristics and insufficient experimental evaluation."*
> *  We admit that our paper is lacking motivations for the proposed heuristics. The intrinsic hyperparameters are searched by try and error until we were able to fit the empirical loss robustly for a ResNet18 and a DenseNet121 trained on Cifar10.
> *  In comparison to (Vaswani et al., 2019) and (Berrada, et al) we did a by far more elaborate evaluation by considering more models and more datasets and at least provide some empirical justifications, whereas (Vaswani et al., 2019) and (Berrada, et al) build their work upon non valid assumptions.
>
>
> *"Despite the method being described as simple and straightforward, I fail to see how the proposed approach is simpler than a backtracking line-search on the batch loss."*
> *  In our opinion the PLS (Mahsereci & Hennig,2017) is closest to our approach. Thus, the straightforward was meant in comparison to PLS. Compared to PLS our approach is very simple and straight forward.
>
>
> *"A main argument for the proposed approach in contrast to previous work is that the minibatch loss is not sufficiently representative for a line-search. This contrasts with observations in previous work (e.g. Vaswani et al., 2019)"*
> *  (Mutschler et al, 2020) showed empirically that (Vaswani et al., 2019) and (Berrada, et al) behave non robust on only slightly different problems. (Vaswani et al., 2019) theoretically show that line-searches on batch losses are applicable if the rather strong interpolation assumption is fullfilled (in addition to convexity and lipschitz continuity). However, they do not show empirically that it is valid for common deep learning tasks. The experiment of (Mutschler et al, 2020) Figure 6 contradicts their assumptions since an optimal line searches on batch losses should actually converge but does not.
>
> *"The experimental evidence in section 3.1 is promising but is not broad enough to be informative. While the results of section 3.2 are nice to have, the improved robustness of the method against SGD and Adam is not surprising; the space would be better used comparing the performance of the method against the other cited works on batch line search."*
> *  A good point. We will also compare against SLS, GOLS-1 and PLS over all considered datasets and models.
>
> *"But I doubt the submission will generate enthusiasm from the theory side of the community do to the limited new insights and the empirical evaluation is insufficient to convince me of the benefits of the method over previous work for its applications."*
> *  We will perform additional experiments which strengthens our empirical foundations.
> *  We will also show how many of the measured lines behave convex on different distances around the minimum to show that the lines are locally convex, this result might interest the theory side of the community since it suggests that there convex assumptions are valid.
>
>
> *"As additional feedback (there to help, not necessarily part of the decision), I advise the authors to pay a greater attention to presentation details when preparing their submissions. The writing and quality of the figures gets in the way of the message they are trying to communicate. For a few examples, since the "homoscedasticity" is irrelevant for the current problem, I do not follow why it is mentioned so prominently in section 2. The use of "vertical cross-section" is also confusing; it is still unclear to me what is vertical."*
>  *  We apologize for this and will do our best to restructure the paper. Due to the detailed feedback we now know well what is unclear and unstructured and will pay attention to it in detail.

---

### Official Review · AnonReviewer3 · 2020-10-28
**Review for ELF**

**Rating:** 4
**Confidence:** 4

**Review:**

Summary:
This work proposes ELF, a newl method to do line search. The key idea is to fit a low order polynomial of the empirical loss (by sampling multiple batches) along the direction of a mini-batch gradient. The method stays computationally efficient by only computing the step size every so often. Experiments on a variety of image classification tasks show that ELF is competitive to GOLS1, PLS, PAL, SGD, and Adam, while taking less time to train.

Strengths:
The fact that ELF can get competitive results compared to SGD with momentum, and Adam, while taking less time to train, is exciting.

Weaknesses:
- I find the contribution of this paper a bit weak. Reading a line search paper with mostly empirical justifications, I would like to see the method outperforming a strong baseline (a commonly used optimizer for that architecture, with default hyperparameters for example) where ELF uses a fixed set of hyperparameters. Such an experiment would justify the practical usefulness of ELF. As far as I can tell, this sort of experiment was not done.
- The paper doesn’t clearly show the role fitting empirical instead of batch loss (which is written as the main contribution) compared to the introduction of beta and delta on performance. How does PAL perform with these additional settings (since they also seem equally applicable)? Similarly, for Figure 3: How much of this is due to the introduction of beta and delta compared to fitting method, or the fact that empirical loss is used rather than batch loss?
- Using the validation set during training doesn’t really make sense. If the method overfits greatly by using just the training set, then that method is prone to overfitting; using the validation set can’t be the solution.
- Currently, the paper’s presentation is lacking. Paragraphs are too long, the related work section feels unorganized, Figure 1 is taken from an already published paper, and the two huge algorithm boxes break the flow of the paper and is too much detail.

Decision:
Reject. The idea shows promise but the current state of the paper should be improved before warranting acceptance, for the reasons listed above.

Comments and questions:
- “Due to CLT we can assume p_data to be Gaussian” - not true.
- Was there a reason why SLS (Vaswani et al. 2019) wasn't included in the comparisons? To my knowledge, SLS is the best performing line search method in deep learning.

---

> ### Author Response · Authors · 2020-11-17
> **Detailed reply to reviewer 3**
>
> **Dear reviewer 3 thanks again for the detailed and good feedback!**
> We hope, that we could answer your questions satisfyingly. As described below we will follow your advice and will redo some experiments and we will add additional ones.
> Note, that there is also a "General reply to all reviewers " above.
>
> *"I find the contribution of this paper a bit weak. Reading a line search paper with mostly empirical justifications, I would like to see the method outperforming a strong baseline (a commonly used optimizer for that architecture, with default hyperparameters for example) where ELF uses a fixed set of hyperparameters. Such an experiment would justify the practical usefulness of ELF. As far as I can tell, this sort of experiment was not done."*
> *  We distance ourselves from this kind of experiments since multiple papers in the line search field for deep learning, such as SLS (Vaswani et al, 2019), ALI-G (Berrada, et al, 2019), L4 (Rolinek et al, 2018), SGD-HD (Bayden et al, 2017) ) show specific experiments in which their approach performs better. However, those results are not valid on a variety of even only slightly different tasks. Please have a look into the evaluation of (Mutschler et al, 2020)!
> In practice people use their own datasets and models, thus showing very strong performance for one specific model trained on one specific dataset is not of interest. Those people just want to have as good results as possible with as less effort as possible (meaning no parameter tuning). Thus, showing that the approach is working robustly over several datasets and models without any hyperparameter tuning is by far more valuable in practice than optimal performance on one specific task.
> However, we indirectly already did what you are asking for:
> The resulting hyperparameters of our robustness check on cifar 10 are identical to the defaulty used hyperparameters. Especially on Mobilenets (Figure 4,8) ELF outperforms the other optimizers. We will include plots of the full training process for these scenarios.
>
> *"The paper doesn’t clearly show the role fitting empirical instead of batch loss (which is written as the main contribution) compared to the introduction of beta and delta on performance. How does PAL perform with these additional settings (since they also seem equally applicable)?"*
> * Our method is basically SGD with a picewise constant learning rate schedule, which automatically estimates when to measure a new learning rate and then automatically measures the new learning rate. Therfore beta is just the beta of SGD with momentum.  We will add an experiment showing the influence of beta and delta in detail! Already now it can be seen in the top left plot of fiugres 4,7,8 that delta increases the performance on test and validation loss and in combination with the momentum term also on training loss.
> PAL already includes a momentum-like term and an update step adaptation which has a similar effect as delta. The influence of those is given in (Mutschler et al, 2020).
>
> *"Similarly, for  Figure 3: How much of this is due to the introduction of beta and delta compared to fitting method, or the fact that empirical loss is used rather than batch loss?"*
> *  We admit that the comparison is biased since for GOLS-I and PLS we did not consider momentum term. We will redo this experiment and will analyze in detail what the influence of beta and delta is.
> *  In addition, we will compare against a line search that will fit the batch loss with lower order polynomials, to emphasize in addition to the results of (Mutschler et al, 2020) that exact line searches on batch losses are not promising.
>
> *"Using the validation set during training doesn’t really make sense"*
>  *  Usually the validation set is introduced to adapt hyperparameters. This is exactly what we are using it for since we only perform line searches every so often to estimate new learning rates. However, we have to admit that this does make plots of validation accuracy over time not comparable. Note, that we used an additional test set!
>
> *"Currently, the paper’s presentation is lacking."*
>   *  We apologize for this and will do our best to restructure the paper. Due to the detailed feedback we now know well what is unclear and unstructured and will pay attention to it in detail.
>
> *"Was there a reason why SLS (Vaswani et al. 2019) wasn't included in the comparisons? To my knowledge, SLS is the best performing line search method in deep learning."*
> *  At leas empirically one could argue that the approach of (Mutschler et al, 2020) outperforms SLS at least in terms of robustness.
> *  (Mutschler et al, 2020)  also showed that SLS is performing not robust on only slightly different optimization problems.
> *  But anyways we will include a comparison to SLS.
>
> *"Due to CLT we can assume p_data to be Gaussian"* - not true.
> *  We apologize for this obvious mistake. What we intended to say is: due to CLT the L_batch is approximately normal distributed with mean L_emp.

---

> > ### Comment · AnonReviewer3 · 2020-11-25
> > **This paper has potential, but not ready at the current stage.**
> >
> > I have read all reviews and the authors’ comments. It seems to me that this paper has a lot of potential, and the proposed changes address most of my concerns. However, at its current state, the paper is far from my standards of acceptance, and therefore, I will not change my score. I will recommend the authors submit to a future conference, with the proposed changes.

---

### Official Review · AnonReviewer1 · 2020-10-29
**Unprincipled line-search approach for optimizing deep neural networks. Needs better comparison to past literature.**

**Rating:** 3
**Confidence:** 5

**Review:**

This paper proposes a line-search for optimizing deep neural networks. The method is rather unprincipled and quite close to the approach proposed in (Vaswani et al, 2019). I do not think that the paper proposes new ideas that haven't been already explored in the deterministic optimization literature. All in all, the paper needs to better connect to the algorithmic ideas in the deterministic optimization literature, compare against the new optimization methods proposed recently and have more representative plots. Detailed review below.

- The claim "adaptive methods concentrate more on finding good directions than on optimal step sizes, and could benefit from line search approaches" needs justification. At this point, there has been substantial work that shows that adaptive methods like Adam are quite robust to their step size and do in fact, work well across problems.

- Please cite the literature relevant to SGD, for example, Robins-Munro and Bottou et al, 2016. In the deterministic optimization literature, using Polyak step size is an alternative to line-search approaches. Please also cite the recently proposed methods based on the Polyak step-size, Berrada, et al "Training Neural Networks for and by Interpolation" and Loizou et al "Stochastic polyak step-size for SGD: An adaptive learning rate for fast convergence" that have been used to train deep neural networks.

- In Figure 1, it is not enough to show that the loss landscapes are similar in the batch gradient direction. they should also be the same in the stochastic gradient direction, which depending on the batch, can be very different from the batch gradient direction. A more convincing approach would be to choose random directions and compute a metric of similarity for all such directions. Moreover, please explain how is "t" chosen for this plot. We know that the loss landscape is different at the start vs the end of the optimization. Consequently, this is not enough evidence to show that line-search can be done on a stochastic batch. In this figure, the batch-size is another confounding factor. What is the effect of the batch-size?

- "lemp has a simple shape and can be approximated well by lower order polynomials, splines or fourier series." This is indeed the motivation behind line-search techniques for convex problems. Please cite Noecedal-Wright 2006 or the original line-search paper by Armijo.

- In Section 2.2, backtracking line-search is used to overcome the first challenge, i.e. the algorithm picks the largest step-size that satisfies a sufficient decrease condition. The complexity of this is proportional to the number of backtracking iterations which is small. In the proposed approach, a number of points is sampled, meaning additional function evaluations. Please justify why the backtracking approach is not sufficient in this case, and explain why the proposed method would be better.

- Similarly, for the second challenge, it is doing exactly the thing you cautioned against earlier "that the batch need not be representative of the full loss function".  Doing a backtracking line-search on a batch is exactly the approach adopted by (Vaswani et al, 2019). Please clarify this and also explain why you do not experimentally compare against them.

- The idea of building a model of the loss function by using additional points (from the past) is well known in the deterministic optimization literature in the form of line-search with quadratic/cubic interpolation. This approach does not use additional function evaluations. Please explicitly make this connection and again, justify your approach against this less expensive method. Moreover, the paper proposes to build a highly accurate model of a stochastic loss which need not be representative of the full loss in any case.

- "The test error is determined by a 5-fold cross-validation. The second last polynomial degree is chosen and the polynomial is again
fitted on all loss values to get a more accurate fit. Consequently the closest minimum to the initial location is determined and additional losses are measured in a reasonable interval around it." This is clearly very computationally expensive for determining the step-size for one iteration. Please clearly state what is the computational complexity of determining the step-size in one iteration.

- "ELF generalizes better if not performing a step to the minimum, but to perform a step that decreased the loss by a decrease factor" This is almost the same as checking the Armijo sufficient decrease condition with a factor of \delta. Why not just do this and say it explicitly?

- Experimentally, since the proposed approach is closest to the work of (Vaswani et al, 2019), please experimentally compare against their method.

- In Figure 4, please plot the training/test loss vs the number of iterations. One point of information in the form of the test accuracy is not representative, especially since the metric being optimized is the training loss. And there are multiple confounding factors that influence the test error corresponding to any optimization method. Since this is more of an experimental paper, it would make sense to compare against the newer variants of Adam, such as RADAM and AdaBound that have found to work well.

---

> ### Author Response · Authors · 2020-11-17
> **Detailed reply to reviewer 1 (Part 1)**
>
> Detailed answer to reviewer 1
> **Dear reviewer 1 again many thanks for the good and unexpectedly detailed feedback!**
> Unfortunately we fear that something in our explanations has led you on the wrong path since our approach is only marginally related to the approach of (Vaswani et al, 2019). We hope that our answers will provide more clarity.
> Note, that there is also a "General reply to all reviewers " above.
>
>
> *"The method is rather unprincipled and quite close to the approach proposed in (Vaswani et al, 2019)"*
> *  In comparison to (Vaswani et al, 2019) our approach might appear unprincipled since we are not relying on very strong assumptions such as convexity and interpolation, which allow the use of very simple algorithms. However, our approach is based on evidences from real world data and does not rely on any assumptions, which are easily proven to be invalid in practice! (We admit, that we assume that our evidences are generally applicable)
> *  Except for the fact that we do a line search, we see no similarity to Vaswani et al, 2019. In detail, we do not follow the interpolation assumption. We perform no line search on batch losses. We perform no backtracking.
>
> *"I do not think that the paper proposes new ideas that haven't been already explored in the deterministic optimization literature"*
> *  To our best knowledge there is no paper trying to fit the expected loss along search directions with Ordinary Least Square Regression. One contribution of our work is the mapping of the stochastic problem to a deterministic one by approximating the expected loss.
>
> *"compare against the new optimization methods proposed recently and have more representative plots"*
> *  We will add additional experiments and will show plots showing metrics over the whole training process.
>
>
> *"The claim "adaptive methods concentrate more on finding good directions than on optimal step sizes, and could benefit from line search approaches" needs justification. At this point, there has been substantial work that shows that adaptive methods like Adam are quite robust to their step size and do in fact, work well across problems"*
> *  Justification: Adaptive methods are estimating directions of lower noise. Commonly they estimate directions by applying specific heuristics to parameter-wise downscale parameters with high noise and upscale parameters with high noise. This is indirectly identical to choosing a specific step size for each parameter. However, in the estimated direction a fixed step size is performed, whereas with the help of line searches a more optimal step can be performed.
>
>
> *"Please cite the literature relevant to SGD, for example, Robins-Munro and Bottou et al, 2016. In the deterministic optimization literature, using Polyak step size is an alternative to line-search approaches. Please also cite the recently proposed methods based on the Polyak step-size, Berrada, et al "Training Neural Networks for and by Interpolation" and Loizou et al "Stochastic polyak step-size for SGD: An adaptive learning rate for fast convergence" that have been used to train deep neural networks."*
>
> *  The Polyak step size provides sufficient decrease for convex! functions. In addition, the minimal function value has to be known. Assuming that it will become 0 is not valid in practical machine learning in which heavy data augmentation is applied which speaks against Loizou et al. And even if it is possible it will lead to severe overfitting. (Mutschler et al, 2020) showed in their empirical results that ALI-G (Berrada, et al) does not perform competitive. But still we see their relevance and will cite those.

---

> > ### Author Response · Authors · 2020-11-17
> > **Detailed reply to reviewer 1 (Part 2)**
> >
> >
> > *"In Figure 1, it is not enough to show that the loss landscapes are similar in the batch gradient direction. they should also be the same in the stochastic gradient direction, which depending on the batch, can be very different from the batch gradient direction. A more convincing approach would be to choose random directions and compute a metric of similarity for all such directions. Moreover, please explain how is "t" chosen for this plot. We know that the loss landscape is different at the start vs the end of the optimization. Consequently, this is not enough evidence to show that line-search can be done on a stochastic batch. In this figure, the batch-size is another confounding factor. What is the effect of the batch-size? "*
> > * Figure 1: For our work we are only interested in the batch gradient direction, since this is the one we use for our line search. What does it help to know the shape of the expected loss function along the expected gradient direction if we cannot estimate it cheaply? But we see the scientific interest in your question.
> > *  Comparing the similarity of search directions originating from different batches at one position in the parameter space and showing that the step sizes are similar is crucial for our paper. We will add such a experiment. Thanks for the hint!
> > *  the plotted cased are consecutive form t = 296 to 301. However, they looked similar for the whole training process. We have more than 50000 of such plots. Thus, they are representative. In addition, we cannot see significant differences on plots form the beginning and the end of the training. We will add plots from the beginning and the end of the training to Figure 1.
> > *   From our experiments we cannot assure that the loss landscape is different at the start vs the end of the optimization. Could you give us references for this? In addition, the results of (Mutschler et al, 2020) which show for thousands of batch loss lines in batch loss gradient directions behave similar over the training (except of for approximately the first 500 steps) This is similarly suggested by (Xing et al. 2018).
> > * You are right that the effect of the batch size is not considered. But, since the batch size is not changing the expected function, changing it will just lead to a higher variance in the batch loss distributions. Considering the loss point clouds we achieved on Imagenet in which the proportion of the batch size to the data set size is much smaller we can still conclude similar shapes. However, using a lower batch size will also increase the variance on the gradient and thus will lead to other search directions. We will perform additional experiments to analyze the effect of the batch size!
> > *  We want to clarify, that only our search direction is stochastic. Although our empirical results are limited, we show by far more empirical evidence than (Berrada, et al) and (Vaswani et al, 2019), which is none. They just rely on strong assumptions.
> >
> >
> > *"In Section 2.2, backtracking line-search is used to overcome the first challenge, i.e. the algorithm picks the largest step-size that satisfies a sufficient decrease condition. The complexity of this is proportional to the number of backtracking iterations which is small. In the proposed approach, a number of points is sampled, meaning additional function evaluations. Please justify why the backtracking approach is not sufficient in this case, and explain why the proposed method would be better. "*
> > * Justification:
> > The crurcial point here is, that SLS (Vaswani et al, 2019) perform each backtracking line search with losses originating from the same batch! However, they can only achieve linear convergence under the rather strong interpolation assumption (in addition to the default assumptions). However, interpolation is not always valid in practice. To our best knowledge they provide no guarantee that the point of sufficient decrease on the batch loss is also valid for the expected empirical loss.
> > In Figure 1 you can see that sufficient decrease on the batch loss might even increase expected empirical loss in practice (However, we have to admit that the expected decrease over multiple steps might be fulfilled).
> > Our approach assures, that we achieve a sufficient decrease condition (in our case the nearest minimum) on the expected empirical loss, which is actually the loss to be optimized. Before we can search for points of sufficient decrease at first we have to approximate expected empirical loss. After knowing the 1D approximation and if it has a simple shape, line searches are not needed anymore because the minimum or a point of sufficient decrease can be determined analytically.
> > Our approach is better since we can ensure sufficient decrease on the expected empirical loss without relying on strong assumptions which are not empirically justified.

---

> > > ### Author Response · Authors · 2020-11-17
> > > **Detailed reply to reviewer 1 (Part 3)**
> > >
> > > *"Similarly, for the second challenge, it is doing exactly the thing you cautioned against earlier "that the batch need not be representative of the full loss function". Doing a backtracking line-search on a batch is exactly the approach adopted by (Vaswani et al, 2019). Please clarify this and also explain why you do not experimentally compare against them"*
> > > *  In contrast to (Vaswani et al, 2019) we only chose the direction from a batch but than estimate the update step on the expected empirical loss. And we are not performing a backtracking line search.
> > > *  The empirical results of (Mutschler et al, 2020) suggest that the performance of (Vaswani et al, 2019) is weak on only slightly different problems and that their approach is not robust.
> > > *  But we see your point and will also compare against SLS.
> > >
> > > "The idea of building a model of the loss function by using additional points (from the past) is wellknown in the deterministic optimization literature in the form of line-search with quadratic/cubic interpolation. This approach does not use additional function evaluations. Please explicitly make this connection and again, justify your approach against this less expensive method. Moreover, the paper proposes to build a highly accurate model of a stochastic loss which need not be representative of the full loss in any case."
> > > *  Determining one loss of the expected empirical loss exactly is more expensive as performing our line search. If using noicy losses interpolations are not valid since continuity is not given. If you look at figure 1 one and you draw 2 or 3 losses out of the random batch distribution at different positions, you will very likely achieve a quadratic or cubic model that does not fit the expected empirical loss at all.
> > > *  Since ordinary least square is unbiased we achieve a exact representation of the full loss if measuring and infinite amount of losses and if the full loss behaves locally like a lower order polynomial. Unfortunately it is unclear to us, what you exactly mean with "stochastic loss" and with "full loss".
> > >
> > > *"lemp has a simple shape and can be approximated well by lower order polynomials, splines or fourrier series." "This is indeed the motivation behind line-search techniques for convex problem"*
> > > We will cite Noecedal-Wright 2006. However, they usually use fixed polynomial degrees <4 ,whereas, our approach actually can consider any degree and chooses the best fitting one.
> > >
> > >
> > > *"This is clearly very computationally expensive for determining the step-size for one iteration. Please clearly state what is the computational complexity of determining the step-size in one iteration."*
> > > *  Firstly, in practice, we perform this line search quite rarely. Applying it at every iteration is practically too expensive.
> > > Our experiments show, that due to better performing step sizes and loaded batches for which no gradient has to be calculated ELF's empirical measured time complexity is lower (see Figure 6 right).
> > > *  if one would perform the line search on each step the instead of one forward pass and one backward pass through the 	network  499 additional forward passes are applied.
> > >
> > >
> > > *"ELF generalizes better if not performing a step to the minimum, but to perform a step that decreased the loss by a decrease factor" This is almost the same as checking the Armijo sufficient decrease condition with a factor of \delta. Why not just do this and say it explicitly?"*
> > > *  Estimating points of a specific value on the approximated one dimensional expected loss is cheap. It would be by far more expensive to start to check positions for the Armijo sufficient decrease condition. But we could solve for a point of sufficient decrease, which is exactly what we are doing.
> > >
> > > *"Experimentally, since the proposed approach is closest to the work of (Vaswani et al, 2019), please experimentally compare against their method."*
> > > * This is not true, our work is much closer to (Mahsereci & Hennig,2017)
> > > ) since they approximate the expected empirical loss with third order polynomial splines and to PAL (Mutschler et al, 2020), which uses second order polynomial approximation on batch losses.

---

> > > > ### Author Response · Authors · 2020-11-17
> > > > **Detailed reply to reviewer 1 (Part 4)**
> > > >
> > > > "In Figure 4, please plot the training/test loss vs the number of iterations. One point of information in the form of the test accuracy is not representative, especially since the metric being optimized is the training loss. And there are multiple confounding factors that influence the test error corresponding to any optimization method. Since this is more of an experimental paper, it would make sense to compare against the newer variants of Adam, such as RADAM and AdaBound that have found to work well."
> > > >  *  The corresponding figures showing the training loss is given in appendix A. We totally agree that showing only the test error is not enough, and should better not be done cause to cofounders. However, it became quite a standard to compare on the test loss. Other reviewers would complain. Radam and AdaBound show similar performance than ADAM (Schmidt et al., 2020). Since Adam is more widely used, we see no problem in comparing to Adam.  We will add some representative training and validation loss curves. However, this can not be done for all (>200) of the trained networks. This is why we chose this condensed representation.

---

### Author Response · Authors · 2020-11-17
**General reply to all reviewers**

**Dear reviewers,
first of all we want to thank you for your excellent and unexpectedly detailed feedback!**
Detailed answers to each of your reviews are given as comments to your posts. Please, have a look at them and please comment if you fully disagree with our answer, because we are always happy to learn something!

**In the following we will address general aspects and criticism and describe what we will change and add to our work.**

 *  Since (Vaswani et al., 2019) and (Berrada, et al) are often mentioned we want to add some words from a data scientists perspective:
(Vaswani et al., 2019) and (Berrada, et al) theoretically show that line-searches on batch losses are applicable if the rather strong interpolation assumption is fullfilled (in addition to convexity and lipschitz continuity). Such results are important since with their help one can show convergence for a broad function space. However, one should still check wether these are valid in practice. But, (Vaswani et al., 2019) and (Berrada, et al)  do not show empirically that it is valid for common deep learning tasks. Further on, (Mutschler et al, 2020) showed empirically that (Vaswani et al., 2019) and (Berrada, et al) behave non robust on only slightly different problems and the experiment of (Mutschler et al, 2020) Figure 6 contradicts their assumptions since an optimal line searches on batch losses should actually converge but does not.
We emphasize this since our approach is based on evidences from real world data and does not rely on their assumptions which are easily proven to be invalid. This is, in our opinion just as valuable as the mathematical approach. We hope, that this argumentation also clarifies the motivation for our work. ( Note, that there is now an accepted and updated version of (Mutschler et al, 2020).)

*  It is often mentioned that the approach of (Vaswani et al., 2019) is similar to our approach.
We do not see this relation. Since we are not performing a backtracking line search, and we are not searching on batch losses.
In our opinion our approach is much closer related to Probabilistic Line Search (Mahsereci & Hennig, 2017) since they also use regressions to approximate the expected empirical loss but from a  probabilistic perspective.

*  We apologize for the unstructured and sometimes confusing and will do our best to restructure the paper. Due to the detailed feedback we now know well what is unclear and unstructured and will pay attention to it in detail.

**We will do the following additions and changes to our work:**
*  We will compare to SLS (Vaswani et al., 2019), ALI-G (Berrada, et al), GOLS-I (Kafka & Wilke, 2019) and PLS (Mahsereci & Hennig, 2017) across all datasets and models we have considered till now. In addition, we will compare to a approach that fits polynomials to batch losses.
*  To make the results comparable, we will let ELF only use the training set.
*  For experiments where ELF performs much better or much worse than the other approaches we will plot training loss, validation accuracy and step sizes for whole training processes.
*  We will perform several ablation studies considering ELF's hyperparameter beta and delta, the amount of losses to be sampled per approximation and the batch size.
*  We will add plots from the beginning and the end of the training to Figure 1. In addition, we will measure the similarity of the expected loss along different lines by a reasonable metric. We will also consider the similarity of the loss along lines in directions originating from different batches measured at the same position in parameter space. We will also quantify how well the expected loss along those lines can be fitted by polynomials.
*  We will restructure the paper in detail.
*  We will clarify the motivation of our work.

*Do you have anything to add, which we missed?*

Since this is a single person project we (I) won't be able to perform all the described additions and changes in the following week. But I promise that they are done as fast as possible.

---

### Decision · Program_Chairs · 2021-01-07
**Final Decision**

**Decision:**

Reject

**Comment:**

This paper proposes an empirical method to automatically schedule the learning rate in stochastic optimization methods for deep learning, based on line-search over a locally fitted model. The idea looks interesting and promising, but the reviewers have serious concerns in the lack of principled support and insufficient empirical evidences. Therefore I recommend rejection of the paper and encourage the author(s) to strengthen the idea and contribution with further theoretical and empirical study.